# Clinical Management of Hypertension, Inflammation and Thrombosis in Hospitalized COVID-19 Patients: Impact on Survival and Concerns

**DOI:** 10.3390/jcm10051073

**Published:** 2021-03-04

**Authors:** Patricia Martínez-Botía, Ángel Bernardo, Andrea Acebes-Huerta, Alberto Caro, Blanca Leoz, Daniel Martínez-Carballeira, Carmen Palomo-Antequera, Inmaculada Soto, Laura Gutiérrez

**Affiliations:** 1Platelet Research Lab, Instituto de Investigación Sanitaria del Principado de Asturias (ISPA), 33011 Oviedo, Spain; uo266133@uniovi.es (P.M.-B.); anbermed@gmail.com (Á.B.); andreaacebeshuerta@gmail.com (A.A.-H.); bercamez@gmail.com (A.C.); daniel_mc@hotmail.es (D.M.-C.); 2Clinical Diagnosis Laboratory, Department of Hematology, Hospital Universitario Central de Asturias (HUCA), 33011 Oviedo, Spain; 3Department of Hematology, Hemostasis Section, Hospital Universitario Central de Asturias (HUCA), 33011 Oviedo, Spain; 4Department of Intensive Medicine, Hospital Universitario Central de Asturias (HUCA), 33011 Oviedo, Spain; blanca.leoz@gmail.com; 5Translational Microbiology Group, Instituto de Investigación Sanitaria del Principado de Asturias (ISPA), 33011 Oviedo, Spain; 6Department of Internal Medicine, Hospital Universitario Central de Asturias (HUCA), 33011 Oviedo, Spain; cpalomoantequera@gmail.com; 7Bone Metabolism, Vascular Metabolism and Chronic Inflammatory Diseases Group, Instituto de Investigación Sanitaria del Principado de Asturias (ISPA), 33011 Oviedo, Spain; 8Department of Medicine, Faculty of Medicine, University of Oviedo, 33006 Oviedo, Spain

**Keywords:** COVID-19, clinical manifestation, comorbidity, thrombosis, corticosteroids, antihypertensive agents

## Abstract

The most severe clinical manifestations of the Coronavirus disease 2019 (COVID-19), caused by severe acute respiratory syndrome coronavirus 2 (SARS-CoV-2), are due to an unbalanced immune response and a pro-thrombotic hemostatic disturbance, with arterial hypertension or diabetes as acknowledged risk factors. While waiting for a specific treatment, the clinical management of hospitalized patients is still a matter of debate, and the effectiveness of treatments to manage clinical manifestations and comorbidities has been questioned. In this study, we aim to assess the impact of the clinical management of arterial hypertension, inflammation and thrombosis on the survival of COVID-19 patients. The Spanish cohorts included in this observational retrospective study are from HM Hospitales (2035 patients) and from Hospital Universitario Central de Asturias (72 patients). Kaplan Meier survival curves, Cox regression and propensity score matching analyses were employed, considering demographic variables, comorbidities and treatment arms (when opportune) as covariates. The management of arterial hypertension with angiotensin-converting enzyme 2 (ACE2) inhibitors or angiotensin receptor blockers is not detrimental, as was initially reported, and neither was the use of non-steroidal anti-inflammatory drugs (NSAIDs). On the contrary, our analysis shows that the use on itself of corticosteroids is not beneficial. Importantly, the management of COVID-19 patients with low molecular weight heparin (LMWH) as an anticoagulant significantly improves the survival of hospitalized patients. These results delineate the current treatment options under debate, supporting the effectiveness of thrombosis prophylaxis on COVID-19 patients as a first-line treatment without the need for compromising the treatment of comorbidities, while suggesting cautiousness when administering corticosteroids.

## 1. Introduction

Coronavirus disease 2019 (COVID-19) is a viral illness caused by severe acute respiratory syndrome coronavirus 2 (SARS-CoV-2), identified in Wuhan (China) in December 2019 and declared a pandemic by the World Health Organization in March 2020 [1]. The absence of prior immunity against this novel type of coronavirus and the lack of specific treatments translated to millions of infected people and thousands of deaths worldwide.

The SARS-CoV-2 spike (S) protein is responsible for facilitating the virus’s entry into the target cells through recognition of the angiotensin-converting enzyme 2 (ACE2) receptor. ACE2 is highly expressed in the lung’s type II alveolar cells, cardiac myocytes, cholangiocytes and in lower amounts in hepatocytes in the liver, as well as the proximal tubule cells of the kidney, bladder urothelial cells and enterocytes of the small intestine, among others, while being weakly expressed on the surface of epithelial cells in the oral and nasal mucosa and nasopharynx [2,3,4]. In most cases, the immune response is self-competent, leading to recovery. However, in some patients, the immune response is unbalanced and non-competent, with age, gender and comorbidities such as arterial hypertension or diabetes being acknowledged risk factors. As a consequence, these patients require hospitalization, with various levels of clinical manifestations that need to be managed appropriately. Respiratory distress in the form of bilateral pneumonia was highlighted as the main adverse clinical manifestation [5]. However, as the pandemic advances, we have learned the systemic nature of the disease, which affects multiple organs and is accompanied by thrombotic events (which may occur in infected patients even post-recovery) [6,7,8]. Poor prognoses in COVID-19 patients are associated with the dysfunctional immune response and concomitant cytokine storm, governing the systemic inflammation and related tissue damage, which occurs with the subjacent contribution of a hyperreactive hemostatic system, ultimately responsible for the thrombotic nature of multi-organ failure [2,9,10,11,12,13,14,15]. The complexity does not end here, especially considering the implications of comorbidities and the crossroads of clinical manifestations of the disease (inflammation and thrombosis) with the renin–angiotensin system at the onset and through disease progression [16,17,18]. Since the start of the pandemic, a large number of reports claiming deleterious, beneficial or innocuous effects of different treatment options have been published.

In particular, the administration of angiotensin-converting enzyme 2 inhibitors (ACE2-Is) or angiotensin receptor blockers (ARBs) on patients suffering from arterial hypertension (AHT) has been questioned [19,20,21,22]. However, we have to consider that poor management of AHT can give rise to thrombotic and bleeding events that may be fatal in COVID-19, acknowledging its pro-thrombotic nature [23].

Considering the alleviation of the inflammatory response and concomitant tissue damage, anti-inflammatory drugs (non-steroidal anti-inflammatory drugs (NSAIDs) or corticosteroids) are being administered to COVID-19 patients with different treatment regimens [24,25]. However, there is controversy regarding their use [26,27]. As an example, ibuprofen has received bad publicity, as it was hypothesized that its administration would result in overexpression of ACE2, which would in turn increase the risk of cell entry by the virus. Even today, despite a number of manuscripts disproving this hypothesis, paracetamol is prioritized as an antipyretic over ibuprofen [28,29,30,31]. Regarding the use of corticosteroids, a recent study showed that low-dose dexamethasone, especially in severely ill COVID-19 patients (i.e., intensive care unit (ICU)-hospitalized patients with respiratory distress), greatly improved their survival [32]. However, the publicity around this publication and the lost information in press releases has contributed to the generalized use of corticosteroids in many hospitals, which may not always be beneficial to a patient not requiring this intervention, as they may disturb or further unbalance an already non-competent immune response [33,34,35].

Management of the hemostatic response is very relevant in COVID-19 patients, due to the pro-thrombotic character of the disease. Anticoagulants (low molecular weight heparin (LMWH) being among them) were initially administered only to those patients presenting thrombotic events or those either immobilized or ICU-hospitalized. Gradually, with the generated knowledge and pandemic progression, several hospitals implemented thromboprophylaxis to all COVID-19 patients immediately upon admission and even after hospital discharge, to prevent complications and thrombotic events that have been reported as a common sequel to the infection. In fact, the term immunothrombosis has been established to refer to the thrombosis of septic shock and acute immune responses due to infections of different etiology [17]. To date, the effectiveness of therapeutic LMWH has been emphatically studied in ICU-hospitalized patients, and information on non-ICU hospitalized or prophylactic approaches in COVID-19 patients is lacking.

Facing COVID-19, a new disease for which a specific treatment has not yet been developed [36,37], it is important to critically evaluate the appropriate treatment to ameliorate a specific clinical manifestation to sensitively improve the survival of hospitalized patients, a key priority given the unprecedented strain on the health systems of most countries. It is of the utmost importance to distinguish the general management of manifestations (which would be maintained whatever the subjacent cause is) and management of manifestations that will aid in combating the infection.

In the present study, we have analyzed real-world data from hospitalized COVID-19 patients (HM Hospitales, Spain) and an independent cohort from our local central university hospital (HUCA, Oviedo, Spain) in order to evaluate the impact of treating the primary clinical manifestations of the disease (thrombosis and inflammation) and the management of AHT as a frequent comorbidity (as explained above) on the survival of COVID-19 patients. Our aim is to critically review the current assumptions regarding patient treatment and to establish priorities for the management of COVID-19 patients, considering, above all, that with no specific treatment available, a risk–benefit rationale must be applied to each patient. Our hypothesis is that the treatment and appropriate management of thrombosis in COVID-19 patients will allow more time for the patient’s own immune system to cope with the infection, resulting in increased survival among hospitalized COVID-19 patients.

## 2. Materials and Methods

### 2.1. Patient Data Sources

Access to the COVID Data Saves Lives dataset, belonging to the HM Hospitales network of 17 hospitals in Spain, was obtained. This dataset is an anonymized registry that contains extensive clinical information of laboratory-confirmed hospitalized COVID-19 adult patients. Appendix A contain the HM cohort data used for this study, with patients registered between 26 December 2019 and 23 April 2020. HM Hospitales makes this clinical dataset available to researchers from academic, university and healthcare institutions who request it and whose projects are approved. The content is expected to be expanded and updated periodically, and its update will not be completed until this pandemic is terminated. To obtain the data, it will be necessary to send the following request to the coviddatasavelives@hmhospitales.com or data_science@hmhospitales.com emails in order to be evaluated by the Data Science Commission and, where appropriate, by the research ethics committee of HM Hospitales or any other accredited research ethics committee.

Anonymized data from an independent cohort of ICU-hospitalized COVID-19 patients recruited from 29 February to 26 May 2020 was obtained from the Central University Hospital of Asturias (HUCA, Oviedo, Spain). Appendix A contain the HUCA cohort data used for this study.

The study was performed following the Declaration of Helsinki regulations and guidelines, abiding the requirements for anonymization of registries.

### 2.2. Variables

The following variables were selected: age, sex, date of general admission, date of ICU admission (if any), date of discharge or death (exitus), diagnosis of diabetes, AHT and registry of thrombotic and bleeding events. Regarding medication, the following variables were included: the administration and dose (prophylactic, intermediate and therapeutic) of low molecular weight heparins (enoxaparin and bemiparin) or fondaparinux (synthetic heparinoid drug), which we will refer to together as LMWHs; anticoagulants (vitamin K antagonists and direct oral anticoagulants) or anti-platelet drugs (clopidogrel, acetylsalicylic acid, triflusal, prasugrel and ticagrelor); corticosteroids (dexamethasone, budesonide, betamethasone, methylprednisolone, clobetasol, hydrocortisone, prednisolone, prednisone and deflazacort); non-steroidal anti-inflammatory drugs (NSAIDs) (ibuprofen, naproxen, dexketoprofen, diclofenac and indomethacin); other immunosuppressing drugs (interferon beta-1b, ciclosporin, mycophenolic acid and azathioprine); and angiotensin-converting enzyme 2 inhibitors (ACE2-Is) (enalapril and captopril) and angiotensin receptor blockers (ARBs) (valsartan), which we will refer to together as anti-AHT drugs.

### 2.3. Study Design

The primary end point was the time from hospital admission to discharge or death. Patients that remained hospitalized at the time of data collection were excluded from the study. Patients under 18 years old or with a hospitalization length beyond 50 days were excluded.

The study was performed as indicated per subsection to assess the impact of different treatment arms (see also Appendix A):AHT: patients included were treated or not treated with ACE2-Is (enalapril or captopril), ARBs (valsartan) or combinations of both ACE2-Is and ARBs. Separate analyses were performed for each one of these subgroups, and they were grouped for all patients receiving ACE2-Is or ARBs;Inflammation (NSAIDs): patients included were treated or not treated with NSAIDs. Patients treated with immunosuppressing drugs or corticosteroids were excluded;Inflammation (corticosteroids): patients included were treated or not treated with corticosteroids. Patients treated with other immunosuppressing drugs (interferon beta-1b, ciclosporin, mycophenolic acid and azathioprine) or NSAIDs were excluded;Hemostasis: patients included were treated or not treated with low molecular weight heparin (LMWH). Patients treated with other anticoagulants (vitamin K antagonists and direct oral anticoagulants) or anti-platelet drugs (clopidogrel, acetylsalicylic acid, triflusal, prasugrel and ticagrelor) were excluded, as they were represented in low numbers and precluded a rigorous statistical analysis. Patients receiving different doses of LMWH during their hospitalization were included in the group of the highest administered dose.

### 2.4. Statistical Analysis

Cox proportional-hazards regression models were used to compare survival rates between the treated and non-treated groups. An initial multivariate Cox regression was performed, which included demographic factors (sex and age), diabetes and AHT as covariates and treatment arms. Additionally, to further control for potential baseline confounding factors across groups and the non-randomized treatment administration, we used a 1:1 pair propensity score matching (PSM) analysis. The individual propensities were estimated with the use of a multivariable logistic regression model, which included those variables that could be affecting both the outcome and the likelihood of receiving that medication (in this case, the same covariates as the Cox regression model: sex, age, diabetes and AHT), and the remaining respective treatment arms as indicated in each subsection of the Methods and Results sections, and the optimal method was applied to create a matched database. The results are presented in the form of Kaplan–Meier survival curves and forest plots with hazard ratios (HRs) and 95% confidence intervals (CIs) for the Cox regression analysis. Statistical significance was considered with a *p*-value < 0.05. All analyses and figures were generated using the survival, survminer, eulerr, forestmodel, optmacht and MatchIt packages in R (version 3.6.1) (R Project for Statistical Computing, Vienna, Austria) [38].

## 3. Results and Discussion

The filtered HM patient cohort consisted of a total of 2035 hospitalized patients, of which 1702 were discharged and 333 died (see Appendix A). The survival probability (SP) at 1 week was 88.1%, and it was 75.1% at 2 week, reaching 50% at 26 days (Figure 1a). The median in-hospital length of stay was 7 days, which was maintained in discharged patients and was 6 days for those who died (Figure 1b and Appendix A). The distribution of drug administration corresponding to the main treatment arms of study (i.e., arterial hypertension (AHT) (ACE2-Is and ARBs, i.e., anti-AHT drugs), inflammation (corticosteroids) and hemostasis (LMWH)) in discharged patients and patients who died (exitus) is represented in Figure 1c and summarized in Appendix A. Cox proportional-hazards regression identified age (older), sex (males) and AHT (but not diabetes) as risk variables (Figure 1d). Among the treatment arms of study, all treatment regimens, including low molecular weight heparin (LMWH) alone or in combination with corticosteroids and anti-AHT drugs, conferred a significant advantage, as opposed to corticosteroids and anti-AHT drugs alone or in combination, having the group of patients not treated with these treatment arms as the control group (Figure 1d).

Given the previous tendency to focus treatment response studies on ICU-hospitalized patients, we decided to stratify the cohort based on the ICU and non-ICU hospitalization of COVID-19 patients. The survival probability (SP) reached 50% at 23 days in ICU-hospitalized patients and 25 days in those hospitalized in general wards (Appendix A). The median stay length of the non-ICU hospitalized patients was 7 days (discharge 7 days and exitus 5 days), while those requiring ICU hospitalization stayed in a hospital a median of 12 days (discharge 14 days and exitus 11.5 days) (Appendix A). While the stay length was sensitively increased in the ICU-hospitalized patients, the SP was only notably different between 10–25 days of hospitalization. The distribution of treatment arms in the stratified cohort based on ICU and non-ICU hospitalization is represented in Appendix A. Cox regression analysis on the non-ICU hospitalized patient group suggested a significant advantage for LMWH-treated patients (Appendix A), with age and sex as maintained risk variables, while comorbidities (AHT and diabetes) did not reach significance. The same analysis on the ICU-hospitalized group did not converge due to the limited number of untreated patients.

In order to evaluate whether ICU-hospitalization would imply a risk on its own, we decided to perform a propensity score matching (PSM) analysis, having sex, age, comorbidities (AHT and diabetes) and treatment arms (all combinations) as covariates. As shown in Appendix A, the 50% SP of the matched groups was the same as the unmatched cohort, considering that the SP was now notably different between 5–25 days of hospitalization. Nevertheless, Cox regression analysis of the matched populations showed no significant hazard derived from an ICU stay in ICU-hospitalized patients compared with patients hospitalized in general wards. For this reason, we continued our analyses without ICU and non-ICU stratification.

### 3.1. Arterial Hypertension

We next examined whether the management of AHT with ACE2 inhibitors and angiotensin receptor blockers (ACE2-Is and ARBs, two anti-AHT drugs) would be detrimental to the SP of COVID-19 patients, as was initially reported and is still controversial [39,40]. The median stay length of the patients treated with anti-AHT drugs was 8 days (7 days non-ICU hospitalized and 16 days ICU hospitalized) (Figure 1b and Appendix A). After PSM analysis with sex, age, comorbidities (AHT and diabetes) and the remaining treatment arms (corticosteroids and LMWH) and their combinations as covariates, we did not observe a reduced SP in COVID-19 patients when treated with anti-AHT drugs, compared with patients that did not receive such a treatment regime (Figure 2a). Rather we observed the contrary. The SP of the matched populations reached 50% at 24 days for patients treated with anti-AHT drugs, while in non-treated patients, a 50% SP was reached at 19 days. Cox regression analysis of the matched populations confirmed that, if anything, the administration of anti-AHT drugs was not detrimental, as has been asserted by others, and might even be beneficial (Figure 2a) [16,19,20,21,39,40,41]. The individual analysis of patients receiving either ACE2-Is, ARBs or a combination of both (Figure 2b) suggested a subtle advantage of ACE2-Is over ARBs and supported the main observation (i.e., the administration of these anti-AHT drugs is not detrimental). Of note, there were a number of patients receiving both anti-AHT drugs in combination which followed the same trend, especially considering the first two weeks of hospitalization. However, due to the low number of patients, the Cox regression analysis did not reach significance.

### 3.2. Inflammation: NSAIDs

Next, we examined the impact of anti-inflammatory drugs (NSAIDs) on the survival of COVID-19 patients, since they were reported to be detrimental (specially ibuprofen), and even as of today, paracetamol is prioritized as an antipyretic compared with ibuprofen, despite contradictory reports [42,43,44]. For this purpose, we filtered out patients treated with corticosteroids and other immunosuppressing drugs (Appendix A). The median stay length of patients treated with NSAIDs was 6 days (Appendix A). After PSM analysis with sex, age, comorbidities (AHT and diabetes) and the remaining treatment arms (anti-AHT drugs and LMWH) and their combinations as covariates, we did not observe a reduced SP in COVID-19 patients when treated with NSAIDs, compared with patients that did not receive them (Figure 2c). The SP of the matched populations reached 50% at 25 days in both groups. Cox regression analysis of the matched populations confirmed that the administration of NSAIDs was not detrimental (Figure 2c).

### 3.3. Inflammation: Corticosteroids

The use of corticosteroids has received many positive reviews, in particular dexamethasone, as treatment for COVID-19 patients, popularly acquiring the qualification as the ultimate cure for the disease, while those scientific reports critically reviewing the results did not get the required dissemination [45,46]. We assessed the impact of corticosteroid administration to COVID-19 patients in the cohort of the study, filtering out those patients treated with NSAIDs or other immunosuppressing drugs (see Appendix A). The patients treated with corticosteroids stayed in the hospital a median of 8 days (6 days non-ICU hospitalized and 10 days ICU hospitalized) (Figure 1 and Appendix A). We performed PSM analysis, having sex, age, comorbidities (AHT and diabetes) and the remaining treatment arms (anti-AHT drugs and LMWH) and their combinations as covariates. The SP of the matched populations reached 50% at 25 days in the group treated with corticosteroids, while it reached 50% SP at 19 days in the non-treated group. However, Cox regression analysis of the matched populations showed that the administration of corticosteroids did not represent a significant advantage (Figure 3a). Most strikingly, when performing this analysis and considering only patients that had received dexamethasone, the SP of the matched groups reached 50% at 23 days, displaying a window of advantage for the non-treated groups between 2 and 23 days (Figure 3b). The median stay length of this treatment group was 8 days (Appendix A). Cox regression analysis of the matched populations did not show a significant advantage for dexamethasone treatment compared with the non-treated group. In fact, when we looked at the global cohort, before applying PSM, the results were quite concerning, as treatment with dexamethasone showed a significant hazard to COVID-19 patients (Appendix A). While this effect was lost after PSM analysis (Figure 3b), these results opened the debate on whether immunosuppression should be considered cautiously and limited to the situations that require it (i.e., to facilitate pulmonary capacity as part of the general guidelines in patients with respiratory distress), but not as a protocolized guideline to treat hospitalized COVID-19 patients in general, as this seems to happen. The distribution of treatment arms in the stratified cohort based on ICU and non-ICU hospitalization is represented in Appendix A, where it can be seen that the percentage of non-ICU hospitalized patients receiving corticosteroids was around 40% in discharged patients and 60% in patients who died.

An important question remains unanswered as to whether the patients receiving corticosteroids were those presenting severe pulmonary affections, thus compromising the previous analysis. In order to overcome this issue, we decided to categorize the patients based on registered diagnosis or not of severe respiratory distress. Patients with respiratory insufficiency, respiratory distress, pulmonary interstitial disease, chronic pulmonary obstruction, asthma, atelectasis, emphysema and pleural edema, which fall under classifications J43, J44, J45, J80, J84, J90, J96 and J98 in the eCIE10ES, were identified, and we performed the same analysis in comparison with patients without registration of those clinical manifestations. Validating this categorization, the Cox regression analysis demonstrated that severe respiratory distress was a significant risk factor (Appendix A). As shown in Table 1, 793 patients had severe respiratory distress (50% were treated with corticosteroids), and 1242 patients did not present severe respiratory distress (of which 44% were treated with corticosteroids). Remarkably, the exitus rate doubled in both categories when comparing the corticosteroid treatment arm with the rest of the patients. Interestingly, the SP of the matched populations reached 50% earlier in the group treated with corticosteroids than in the non-treated group, regardless of the presentation or lack of presentation of severe respiratory distress (Figure 3c). As observed in the global analysis (Figure 3a), Cox regression analysis of the matched populations showed that the administration of corticosteroids did not represent a significant advantage in either category (Figure 3c). The same trend was observed when studying dexamethasone alone, although the patient numbers were very low, and the SPs could not be calculated (Figure 3d).

We should consider that corticosteroids in particular may unbalance the immune response in unexpected ways, which might compromise the fight against the virus [33,34,45,46]. Paradoxically, the use of steroids can potentially cause an unbalanced immune response (e.g., lower cytokine release, but enhanced cytokine-receptor expression plus neutrophil mobilization increase, among others), which may not favor good evolution in these patients [47,48].

### 3.4. Hemostasis

We next set out to study the impact of anticoagulation with LMWH on the survival of COVID-19 patients. We filtered out patients receiving oral anticoagulants and anti-platelet drugs, as their representation was low for performing statistical analyses, and because oral anticoagulants are generally replaced by LMWH in COVID-19 hospitalized patients due to incompatibilities with other treatments or compromised tolerability due to respiratory affections or intubation. The median stay length of patients treated with LMWH was 7 days (7 days non-ICU hospitalized and 12 days ICU hospitalized) (Figure 1 and Appendix A). After PSM analysis with sex, age, comorbidities (AHT and diabetes) and the remaining treatment arms (corticosteroids and anti-AHT drugs) and their combinations as covariates, we did observe an improved SP for COVID-19 patients when treated with LMWH compared with patients that did not receive such a treatment regime (Figure 4a). The SP of the matched groups reached 50% at 24 days in LMWH-treated patients, while it reached 50% at 19 days in the patient group not treated with LMWH. Cox regression analysis of the matched groups showed a significant advantage for the group treated with LMWH (Figure 4a).

We have to consider that, at first, LMWH was administered only therapeutically when severe thrombotic events manifested. As the pandemic advanced, the tendency to implement the prophylactic administration of LMWH became more popular, although this measure has not been protocolized globally [49]. The median stay length of patients treated with prophylactic doses of LMWH was 7 days, while those treated with therapeutic doses of LMWH stayed in a hospital a median of 9 days (Appendix A).

Prophylactic administration of LMWH was associated with a reduced number of thrombotic and bleeding events compared with therapeutic administration (Appendix A). This data suggests that thrombosis prophylaxis might be beneficial to COVID-19 patients and might be associated with improved survival and reduced thrombotic and hemorrhagic events.

We aimed to validate our observations on an independent cohort of patients from our regional hospital, which already implemented the prophylactic use of LMWH as a protocolized measure (see Appendix A). In fact, the SP of these patients (all ICU-hospitalized), which received from the beginning a prophylactic administration of LMWH, was improved compared with ICU patients from HM hospitales. The SP at 1 week and 2 week was 92.7% (Figure 4b), reaching 50% at 39 days (a considerable delay compared with the 23 days observed in the ICU-hospitalized patients of the HM cohort (see Appendix A). HUCA ICU-hospitalized patients presented a median stay length of 15 days (discharge 15 days and exitus 23 days) (Figure 4c).). As shown in Appendix A, bleeding or hemorrhagic events were minimal in the HUCA cohort, which was treated with LMWH in a prophylactic manner. Currently, several clinical trials study the effectiveness of anticoagulants (with different forms of administration (e.g., oral, intravenous, subcutaneous or nebulized) or administered in a therapeutic or prophylactic manner) [9,50,51,52,53,54].

## 4. Conclusions

The management of AHT with anti-AHT drugs did not result in reduced a SP for the treated patients. These results suggest that the treatment of chronic illnesses should not necessarily be interrupted or discontinued without a personalized assessment of the benefit–risk balance. Similarly, it is important to acknowledge that, contrary to what has been previously reported, the use of NSAIDs, while not being beneficial, is also not detrimental to COVID-19 patients. These are examples of how misinformation has led to the assumption of facts, something from which we should learn, as was discussed rigorously for precisely these treatment arms [55].

Importantly, the use of corticosteroids does not exert a beneficial effect on the in-hospital survival of treated patients, and our results suggest that it should be considered as treatment only for those patients requiring it (i.e., in patients with respiratory distress) and not to be implemented as a protocolized approach to treat COVID-19 patients in general. In particular, treatment with dexamethasone may potentially be detrimental to those patients in general wards (not critically ill and without respiratory distress). From our point of view, corticosteroids might elicit an undesired effect on the overall immune response, and more targeted alternatives should be considered such as the containment of tissue infiltration by neutrophils with sivelestat, a neutrophil elastase inhibitor which has been proven effective to inhibit their transmigration capacity [56,57]. Current randomized studies consider the administration of anti-interleukin drugs as a more targeted alternative to modulate the immune response (the COV-AID trial, NCT04330638) [58].

On the other hand, the management of COVID-19 patients with LMWH as an anticoagulant significantly improves the survival of hospitalized patients, and our results suggested that its administration in a prophylactic manner is beneficial, since prevention of thrombotic complications is associated with a better prognosis (improved survival). These results have been independently obtained from the same cohort (HM Hospitales) as previously reported [59] and are supported by a recent pool analysis [60]. Furthermore, the implementation of the use of LMWH upon admission in a prophylactic manner did not associate with increased bleeding, which was also supported by the same study [60]. Current ongoing or planned randomized studies will address the effect of prophylactic or therapeutic LMWH doses in COVID-19 patients (e.g., the IMPACT trial, NCT04406389 (only critically ill patients); the COVID-HEP trial, NCT04345848; the COALIZAO ACTION trial, NCT04394377; the CORIMMUNO-COAG trial, NCT04344756; and the FREEDOM COVID trial, NCT04512079) with increasing demand from the scientific and clinical communities [61,62,63]. Interestingly, it has been reported that heparin inhibits the cellular invasion by SARS-CoV-2 through interaction with the spike protein of the virus, as shown in vitro, using the equivalent to prophylactic doses [64]. A later study showed that unfractionated heparin had stronger antiviral activity in vitro compared with LMWHs [65]. Whether LMWHs may provide an antiviral effect to treated patients beyond thromboprophylaxis needs to be further studied in the clinical setting.

Despite the limitations inherent to observational and retrospective studies performed on real-world data and the need to confirm our findings in larger independent cohorts, this study highlights that the appropriate management of the hemostatic pro-thrombotic nature of the infection in hospitalized patients confers a general advantage to treated patients, while the administration of corticosteroids should be evaluated individually based on clinical needs until a specific treatment for the disease is developed (graphical abstract) [8].

## Figures and Tables

**Figure 1 jcm-10-01073-f001:**
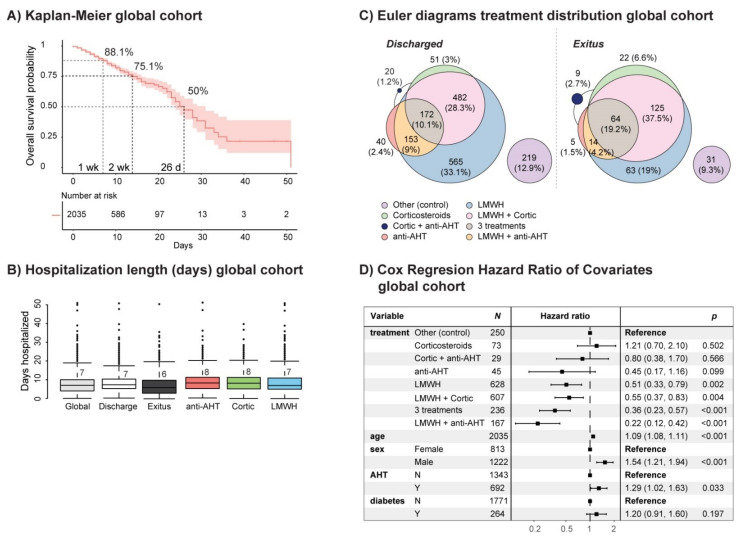
Coronavirus disease 2019 (COVID-19) patient cohort survival rate, hazard ratio of covariates and treatment distribution. (**A**) Kaplan–Meier curve of COVID-19 patients in the overall population of the HM cohort. The red shadow indicates the 95% confidence interval. (**B**) Median stay length of the global HM cohort and groups of study based on discharge or exitus and the treatment arm. (**C**) Euler diagrams displaying the treatment distribution in the COVID-19 patient HM cohort, stratified by discharge or exitus. (**D**) Forest plot showing covariates, obtained after Cox proportional-hazards regression analysis. N, no; Y, yes; wk, week; d, day; LMWH, low molecular weight heparin; AHT, arterial hypertension; Cortic, corticosteroids.

**Figure 2 jcm-10-01073-f002:**
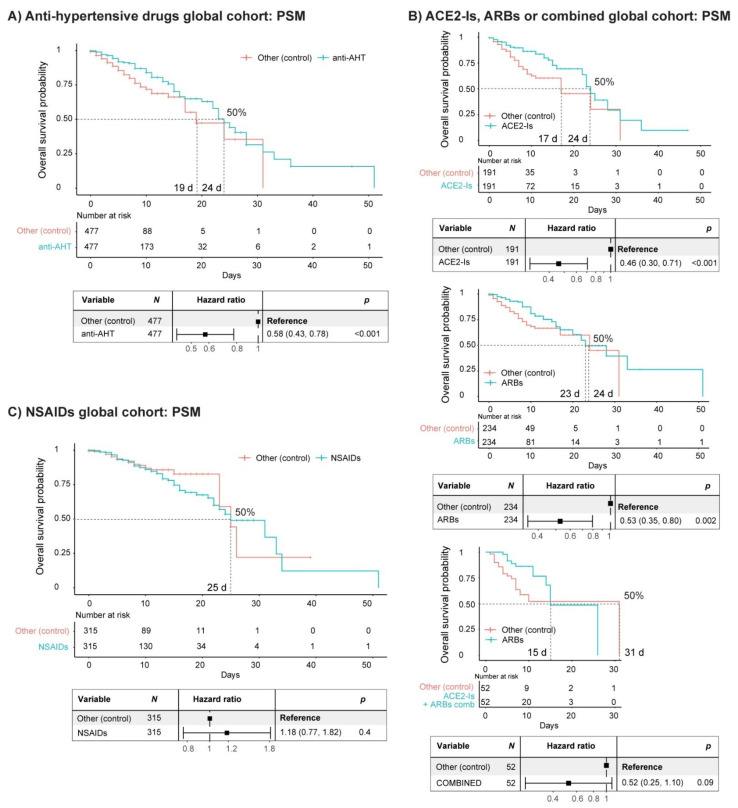
Effect of anti-arterial hypertension (anti-AHT) drugs and non-steroidal anti-inflammatory drugs (NSAIDs) on the survival of COVID-19 patients. (**A**) Kaplan–Meier curve and forest plot obtained from a Cox regression after propensity score matching (PSM) of COVID-19 patients (HM cohort) treated or without anti-AHT drugs. The PSM covariates were sex, age, comorbidities (AHT and diabetes) and treatment arms (corticosteroids and low molecular weight heparin (LMWH)) and their combinations. (**B**) Same analysis as shown in (**A**) after separating patients treated with angiotensin-converting enzyme 2 inhibitors (ACE2-Is), angiotensin receptor blockers (ARBs) or a combination of both (combined). (**C**) Kaplan–Meier curve and forest plot obtained from a Cox regression after PSM of COVID-19 patients (HM cohort) treated or without NSAIDs. Patients treated with corticosteroids or other immunosuppressing drugs were filtered out. The PSM covariates were sex, age, comorbidities (AHT and diabetes) and treatment arms (anti-AHT drugs and LMWH) and their combinations. d, days.

**Figure 3 jcm-10-01073-f003:**
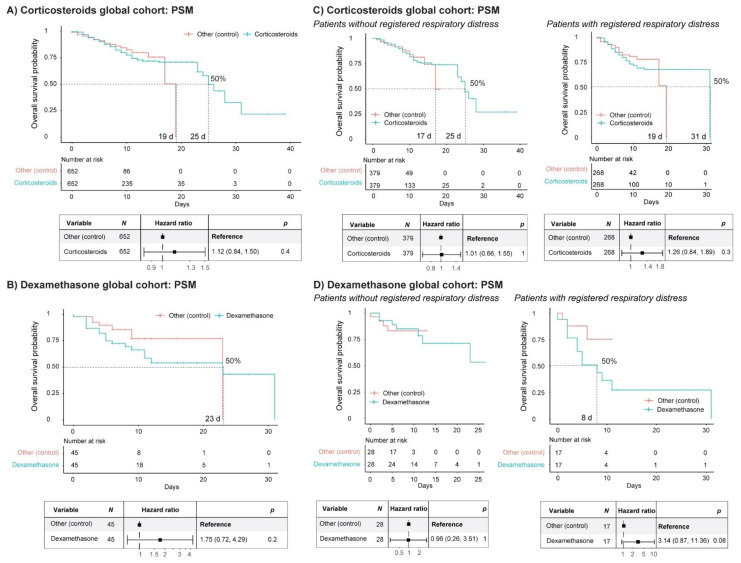
Impact of corticosteroids on the survival of COVID-19 patients. (**A**) Kaplan–Meier curve and forest plot obtained from Cox regression after propensity score matching (PSM) of COVID-19 patients (HM cohort) treated or without corticosteroids. Patients treated with non-steroidal anti-inflammatory drugs (NSAIDs) or other immunosuppressing drugs were filtered out. The PSM covariates were sex, age, comorbidities (arterial hypertension [AHT] and diabetes) and treatment arms (anti-AHT drugs and LMWH) and their combinations. (**B**) Kaplan–Meier curve and forest plot obtained from Cox regression after PSM of COVID-19 patients (HM cohort) treated or without dexamethasone. Patients treated with other corticosteroids (or combinations), NSAIDs or other immunosuppressing drugs were filtered out. The PSM covariates were sex, age, comorbidities (AHT and diabetes) and treatment arms (anti-AHT drugs and LMWH) and their combinations. (**C**) The same analysis as in (**A**) after separating patients with registered respiratory distress from those without registered respiratory distress. (**D**) The same analysis as in (**B**), separating patients with registered respiratory distress from those without registered respiratory distress. d, days.

**Figure 4 jcm-10-01073-f004:**
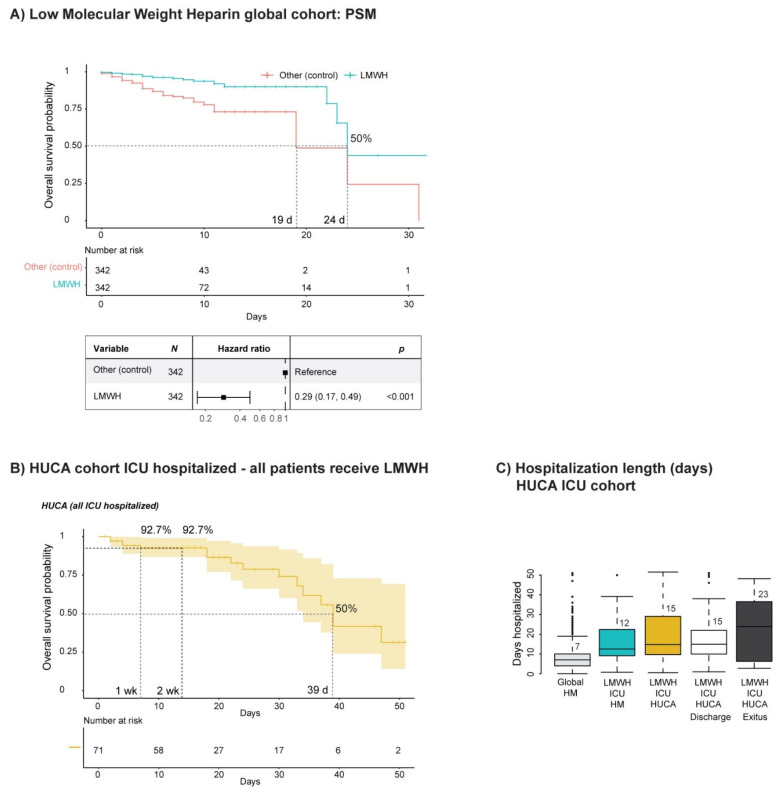
The effect of LMWH administration on the survival of COVID-19 patients. (**A**) Kaplan–Meier curve and forest plot from a Cox regression after propensity score matching (PSM) of COVID-19 patients (HM cohort) that received low molecular weight heparin (LMWH). Patients receiving oral anticoagulants were filtered out. The PSM covariates were sex, age, comorbidities (arterial hypertension [AHT] and diabetes) and treatment arms (corticosteroids and anti-AHT drugs) and their combinations. (**B**) Kaplan–Meier curve of COVID-19 patients (HUCA cohort, ICU-hospitalized) that received LMWH in a prophylactic manner as protocolized. (**C**) Median stay length of COVID-19 patients (HUCA cohort) stratified based on ICU or non-ICU hospitalization, including the median stay of the global and ICU-hospitalized groups from the HM cohort in comparison. d, days; wk, week.

**Table 1 jcm-10-01073-t001:** Distribution of patients after categorization into patients with or without registered severe respiratory distress

**Severe Respiratory Distress** **(*N*)**	**Non-Treated with Corticosteroids** **394 (50%)**	**Treated with Corticosteroids** **399 (50%)**	**Total** **793 (100%)**
**Age**			
*(Mean ± SD)*	67.0 ± 16.1	69.8 ± 14.8	68.4 ± 15.5
**Sex**			
*Female*	170 (43%)	138 (35%)	308 (39%)
*Male*	224 (57%)	261 (65%)	485 (61%)
**Status**			
*Discharge*	334 (85%)	288 (72%)	622 (78%)
*Exitus*	60 (15%)	111 (28%)	171 (22%)
**Hospitalization**			
*Non-ICU*	374 (95%)	349 (87%)	723 (91%)
*ICU*	20 (5%)	50 (13%)	70 (9%)
**No Registered Respiratory Distress** **(*N*)**	**Non-Treated with Corticosteroids** **696 (56%)**	**Treated with Corticosteroids** **546 (44%)**	**Total** **1242 (100%)**
**Age**			
*(Mean ± SD)*	64.2 ± 17.4	68.9 ± 14.7	66.3 ± 16.4
**Sex**			
*Female*	293 (42%)	212 (39%)	505 (41%)
*Male*	403 (58%)	334 (61%)	737 (59%)
**Status**			
*Discharge*	643 (92%)	437 (80%)	1080 (87%)
*Exitus*	53 (8%)	109 (20%)	162 (13%)
**Hospitalization**			
*Non-ICU*	679 (98%)	494 (90%)	1173 (94%)
*ICU*	17 (2%)	52 (10%)	69 (6%)

## Data Availability

The data presented in this study are available in Appendix A. Access to the complete COVID Data Saves Lives dataset, belonging to the HM Hospitales network of 17 hospitals in Spain, can be requested by email to coviddatasavelives@hmhospitales.com or data_science@hmhospitales.com (https://www.hmhospitales.com/coronavirus/covid-data-save-lives/english-version, accessed on 4 March 2021).

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
