# Peer review of "Clinical Management of Hypertension, Inflammation and Thrombosis in Hospitalized COVID-19 Patients: Impact on Survival and Concerns"

_jcm, 2021, doi:10.3390/jcm10051073_

Round 1

Reviewer 1 Report

The analysis of the effect of anti hypertensive drugs on SP is too broad and it was probably more helpful to analyse specific classes

The retrospective nature of the data can not help in propagating a change in treatment protocols

The data should be analyzed in the AHT individually

Author Response

Response Reviewer 1:

We are thankful to the reviewer for the suggested analysis, separating the anti-AHT drugs considered in our study (ACE2-Is and ARBs).

We have now reanalyzed the data shown in Figure 2A, and we have added the analysis on each subgroup of patients as Figure 2B: i.e. patients receiving ACE2-Is (enalapril or captopril), patients receiving ARBs (valsartan), and patients receiving a combination of both ACE2-Is and ARBs. We specify this in the Methods section, pg 4, Line 165 and in the results section, pg 6, line 257.

We agree with the reviewer that the retrospective nature of the data cannot help in propagating a change in treatment protocols. Indeed, this manuscript is not meant as guideline, but aims at reading as a mind-opener, highlighting some misconceptions that should be considered and carefully evaluated in prospective studies (as detailed in pg 11, line 428 for anti-interleukins, and lines 438-441, with new refs 60-62 for LMWHs), with particular emphasis on the beneficial effects of thromboprophylaxis.

Reviewer 2 Report

A very interesting study by a group of scientists from Spain. Overall, the article is written very well. The aim of the work has been clearly defined and the methodology used does not raise any objections.

I am enclosing only minor comments, which I am asking the authors to consider:
1. the introduction is written well and interesting, but I would condense it by highlighting the most important data on COVID-19 relating to the purpose of the paper.
2. 2. I ask the authors to quote, in my opinion, relevant manuscripts in this field:

a. doi: 10.1055/s-0040-1721664
b. doi: 10.3390/pathogens9060493
c. doi: 10.1055/s-0040-1721319

In general, the present article summarizes many important elements of the pathophysiology and treatment of COVID-19, with a very large practical dimension and will certainly be often cited among scientists dealing with this issue.

Author Response

Response Reviewer 2:

We thank the reviewer for the positive review and raised comments.

We have reduced the Introduction by removing some redundant paragraphs, we hope it is clearer and more focused now.

We have referred to the articles suggested by the reviewer, namely new references 8, 59, 63 and 64 (related to 63), added by us.

Reference 8 (doi: 10.1055/s-0040-1721664) is cited in the introduction (pg 2, line 61) and when we link to the now Graphical Asbtract (pg 13, line 462), as it is a good and illustrating review on the systemic manifestations of the disease.

Reference 59 (doi: 10.3390/pathogens9060493) is cited in the Conclusion (pg 11, lines 434 and 436) as it supports our observations regarding the benefits of the treatment with LMWHs through a pool analysis.

Reference 63 (doi: 10.1055/s-0040-1721319) is cited in the Conclusion (pg 13, lines 441-447, together with Reference 64 with the following text “Interestingly, it has been reported that heparin inhibits the cellular invasion by SARS-CoV-2 through the interaction with the Spike protein of the virus, as shown in vitro, using the equivalent to prophylactic doses [63]. A later study has shown that unfractionated heparin has stronger antiviral activity in vitro compared to LMWHs [64]. Whether LMWHs may provide with antiviral effect to treated patients beyond prophylaxis, needs to be further studied in the clinical setting”.

Reviewer 3 Report

The authors describe their experience in a cohort of Covid19 patients admitted to hospital. They draw a myriad of conclusions, none of which are supported by the data presented and in any way possible to draw based on the methodology of this study. The study methodology and interpretation of data is also based on a series of misconceptions. The following comments may be made:  

  1. This is a two-centre study but the contribution from the two centres is very unbalanced with more than 2000 patients from one centre and only 72 from the other. The advantage of a two centre study is the larger external validity, however, with this imbalance that can hardly be the case and it will merely be a confounding factor.

  1. I do not think that this retrospective cohort study is able to conclude that ‘use of corticosteroids is not beneficial’ as in very large randomized controlled studies the use of dexamethason was shown to have a very positive impact on outcome in Covid19 patients. This conclusion from this paper is not based on clear data derived with a solid methodology and is almost dangerous to publish in this way.

  1. Following the previous issue, patients who received corticosteroids were excluded from this analysis (line 178). Hence, impossible to conclude on their efficacy.

  1. This is a descriptive and observational retrospective cohort study that is known to introduce all kinds of bias when comparing outcomes as related to various interventions. Unfortunately, the authors do not even attempt to reduce this by for example propensity matching, which make their conclusions likely not to be valid.

  1. Line 173: why was antihypertensive treatment limited to ACE inhibitors and Angiotensin antagonists? There must have been a lot of patients on beta blockers, calcium antagonists, diuretics and other classes of antihypertensive drugs.

  1. A confusing fact on this paper is that whereas indeed hypertension and diabetes are risk factors for more severe Covid19, the management of these conditions during the illness has not at all been linked to better or worse outcomes. The authors seem to confuse this throughout the paper.

  1. The introduction of this manuscript is excessively long and unfocused. Please try to concentrate on what you want to study, painting the background and supporting your study questions as concise as possible.  

  1. SARS-CoV-2 primarily enters the body through cells expressing ACE2 receptor in the nose, throat and mouth, mist unfortunately missing in the description in line 47.

  1. Line 49: “renal or hepatic cells”: please be more specific.

10. Line 97: to state that anticoagulants are only administered to those with thrombotic events or ICU patients is a gross misconception. According to established international guidelines the vast majority of patients receiving LMW heparin prophylaxis are general ward patients who have impaired mobility.

11. Line 333: the remark that oral anticoagulants are generally replaced by LMW heparin in hospitalized patients is yet another misconception. This is simply incorrect: in the vast majority of patients this therapy is just continued. All the more reason not to exclude these patients.

12. Figure 5 looks nice but really does not make any sense and is certainly not based on any result presented in this manuscript.

Reviewer 4 Report

This paper by Martínez-Botía et al. is a very interesting work, that retrospectively analyzes the effect of some of the principal treatment options for Covid-19. The findings about corticosteroids and LMWH are of particular interest.

I have a few questions and observations:

  • Line 360 you talk about table 1. However, I was not able to find Table 1 in the article. I was only able to find the 2 tables in the supplementary material Table S1 and S2
  • The results about corticosteroids are surprising and might be of great importance. However, as you acknowledge in your manuscript, we have to face the limitations of a retrospective study. In particular, it might be the case that corticosteroids were used in those patients who had more pulmonary involvement; if this assumption were true, it would be difficult to compare the two groups. Is it possible to analyze the pulmonary involvement of patients who were started steroids and those who were not?
  • For the above-mentioned reasons, I would modify the final sentence “appropriate management of the hemostatic pro-thrombotic nature of the infection in hospitalized patients seems more relevant and efficient than hampering the immune response”. I would not write it as a comparison between the effect of heparin and steroids, since in my opinion there are no sound data to make such a comparison

Author Response

We thank the reviewer for the positive review and raised comments, which we think have added value to the manuscript.

The reviewer is right, we seem to have misplaced Table 1 when submitting the article. It has been submitted now separately as Table S1, as it is very large to include in the manuscript format. Hence, previous Tables S1 and S2 are now S2 and S3, respectively.

Regarding the issue on the severity of the pulmonary affection, we find this very relevant. Almost 100% of patients had a respiratory affection of some kind. We have reanalyzed the data in order to take into account those patients with a more severe pulmonary involvement (i.e. with registered diagnosed respiratory distress such as respiratory insufficiency, respiratory distress, pulmonary interstitial disease, chronic pulmonary obstruction, asthma, atelectasis, emphysema, and pleural edema, which fall under classification J43, J44, J45, J80, J84, J90, J96, J98 in the eCIE10ES - https://eciemaps.mscbs.gob.es/ecieMaps/browser/index_10_mc.html -). Of note, corticosteroids were administered indistinctively to patients with registered respiratory distress or not (see new Table 1, pg 9). As shown now in Supplementary Figure 3B, the diagnosis of a severe respiratory distress is a risk factor (as anticipated and validating our selection). We have separated the patient cohort on those with severe respiratory distress from those without that manifestation (Figure 3B for Corticosteroids and Figure 3D for Dexamethasone), and have observed that the use of corticosteroids does not imply advantage for either group of patients, thus corroborating the global cohort results. We discuss this on the results section, pg 8 lines 319-339.

In any case, we have toned down the last sentence (pg 13, lines 456-462) into: “this study highlights that the appropriate management of the hemostatic pro-thrombotic nature of the infection in hospitalized patients confers a general advantage to treated patients, while the administration of corticosteroids should be evaluated individually based on clinical needs, until a specific treatment for the disease is developed (Graphical Abstract)”.

Round 2

Reviewer 3 Report

Unfortunately the authors have not substantially responded to my previous concerns. I think this manuscript does not meet the standard for publication due to methodological shortcomings.